

# Temporal changes in the structure of a plant-frugivore network are influenced by bird migration and fruit availability

Michelle Ramos-Robles[1], Ellen Andresen[2] and Cecilia Díaz-Castelazo[1]

[1] Red de Interacciones Multitróficas, Instituto de Ecología, A. C. Xalapa, Veracruz, México
[2] Instituto de Investigaciones en Ecosistemas y Sustentabilidad, Universidad Nacional Autónoma de México, Morelia, Michoacán, México

## ABSTRACT

**Background.** Ecological communities are dynamic collections whose composition and structure change over time, making up complex interspecific interaction networks. Mutualistic plant–animal networks can be approached through complex network analysis; these networks are characterized by a nested structure consisting of a core of generalist species, which endows the network with stability and robustness against disturbance. Those mutualistic network structures can vary as a consequence of seasonal fluctuations and food availability, as well as the arrival of new species into the system that might disorder the mutualistic network structure (e.g., a decrease in nested pattern). However, there is no assessment on how the arrival of migratory species into seasonal tropical systems can modify such patterns. Emergent and fine structural temporal patterns are adressed here for the first time for plant-frugivorous bird networks in a highly seasonal tropical environment.

**Methods.** In a plant-frugivorous bird community, we analyzed the temporal turnover of bird species comprising the network core and periphery of ten temporal interaction networks resulting from different bird migration periods. Additionally, we evaluated how fruit abundance and richness, as well as the arrival of migratory birds into the system, explained the temporal changes in network parameters such as network size, connectance, nestedness, specialization, interaction strength asymmetry and niche overlap. The analysis included data from 10 quantitative plant-frugivorous bird networks registered from November 2013 to November 2014.

**Results.** We registered a total of 319 interactions between 42 plant species and 44 frugivorous bird species; only ten bird species were part of the network core. We witnessed a noteworthy turnover of the species comprising the network periphery during migration periods, as opposed to the network core, which did not show significant temporal changes in species composition. Our results revealed that migration and fruit richness explain the temporal variations in network size, connectance, nestedness and interaction strength asymmetry. On the other hand, fruit abundance only explained connectance and nestedness.

**Discussion.** By means of a fine-resolution temporal analysis, we evidenced for the first time how temporal changes in the interaction network structure respond to the arrival of migratory species into the system and to fruit availability. Additionally, few migratory bird species are important links for structuring networks, while most of them were peripheral species. We showed the relevance of studying bird–plant interactions at fine temporal scales, considering changing scenarios of species composition with a quantitative network approach.

Corresponding author
Michelle Ramos-Robles,
ramosrobles.m@gmail.com,
michelle.ramos@posgrado.ecologia.edu.mx

## INTRODUCTION

Ecological communities are collections of interacting species, and their composition and structure change over time (*Jordano*, *1987*; *Olesen et al.*, *2008*; *Petanidou et al.*, *2008*). The study of the intricate network of interacting species can be addressed through the analysis of complex networks, where nodes represent different species and links indicate interactions between them. With the recent advance of computational capabilities, the analysis of complex networks is increasingly being used in ecology for quantitative analyses, with particular emphasis on the study of plant–animal interactions (*Bascompte & Jordano*, *2007*; *Bascompte*, *2009*). Unraveling how interactions between plants and animals are structured is crucial for understanding the ecological and evolutionary processes that support ecosystem function and diversity (*Herrera & Pellmyr*, *2002*).

Plant–animal mutualistic networks are characterized by a nested structure (*Bascompte et al.*, *2003*; *Bascompte & Jordano*, *2006*). This pattern implies two fundamental characteristics that are relevant for network function: (1) a group of generalist species (network core) that comprise most interactions in the network and (2) a group of specialist species (network periphery) that have few interactions, mostly with generalist species (*Bascompte & Jordano*, *2006*). Other important parameters of mutualistic networks include network size, connectance, specialization, interaction strength asymmetry, and niche overlap (see 'Methods' for definitions). Several of these characteristics have been shown to provide mutualistic networks with stability and robustness against disturbance (*Rezende, Jordano & Bascompte*, *2007*; *Alarcón, Waser & Ollerton*, *2008*; *Petanidou et al.*, *2008*; *Díaz-Castelazo et al.*, *2010*).

There has been great progress in identifying and describing plant–animal mutualistic networks structural patterns in recent years. The current challenge is to infer which processes are involved in the configuration of such structural patterns and how these may change over time. It is recognized, for example, that temporal changes in network structure and species composition may occur because of seasonal variability in weather or food abundance (*Carciner, Jordano & Melian*, *2009*; *Vázquez et al.*, *2009*; *Rico-Gray et al.*, *2012*). Plant species that produce large amounts of fruit tend to interact with a large number of frugivorous species (*González-Castro et al.*, *2012*). Consequently, plant phenological cycles, particularly in highly seasonal ecosystems, might cause temporal changes in network characteristics, such as specialization patterns and interaction intensity (*Plein et al.*, *2013*; *Mulwa et al.*, *2013*). On the other hand, other variables related to food availability besides fruit abundance, such as the richness of fruiting species, have received less attention in plant-frugivore network studies.

The presence of new species in a system is an additional factor that could be associated with temporal variations in network structure and composition (*Traveset & Richardson*, *2006*; *Aizen, Morales & Morales*, *2008*; *García et al.*, *2014*; *Traveset & Richardson*, *2014*). Incoming species could alter the behavior and resource use patterns of resident species,

introducing disorder into the plant-frugivore network structure (e.g., loss of the nested pattern), which in turn may affect ecosystem function (*Traveset & Richardson*, *2006*; *Traveset & Richardson*, *2014*). The incorporation of new species into a system may be due to anthropogenic disturbances (e.g., introduction of exotic species) as well as natural processes that belong to the systems dynamics (e.g., animal migration).

Several studies have highlighted the importance of migratory species in tropical bird communities (*Hutto*, *1980*; *Stiles*, *1980*), while others emphasize their role in seed dispersal (*Jordano*, *1987*). It is also known that migratory and resident species may show differences in habitat use (*Loiselle*, *1987*), because migratory birds take advantage of resources that residents rarely use, which could lead to lower competition for resources among them (*Terborgh & Diamond*, *1970*; *Leck*, *1972*). Migrants, for example, often use highly abundant resources, as well as those found in patches or open habitats, which is a characteristic that has been associated with lower disturbance susceptibility (*Leck*, *1972*; *Mulwa et al.*, *2013*). The interactions between resident and migratory birds can affect fruit-frugivore networks, especially in highly seasonal tropical ecosystems (*Loiselle & Blake*, *1991*; *Poulin, Lefebvre & McNeil*, *1992*; *Poulin, Lefebvre & McNeil*, *1993*), and particularly at small spatial scales (*Rey*, *1995*).

Network structural patterns can be studied at different spatial and temporal scales (*Carciner, Jordano & Melian*, *2009*). Short-term (i.e., monthly, seasonal) changes in resource availability or community composition may in turn cause short-term changes in interaction patterns (*Herrera*, *1984*; *Jordano*, *1994*; *Vázquez et al.*, *2007*; *Carciner, Jordano & Melian*, *2009*). To our knowledge, only one study has used network analysis to assess the effect of migrants on structural attributes of a plant-frugivore network, finding that the nested pattern was not affected (*García-Quintas & Parada*, *2014*). Our study focuses on short-term temporal changes in network patterns, which we believe to be particularly relevant for highly seasonal tropical forests, as both fruit availability and bird assemblage composition may be highly variable at this temporal scale in these ecosystems.

We present the first study that evaluates how the proportion of migratory frugivorous bird species, as well as changes in food availability (considering both fruit richness and fruit abundance), affect the structure of plant-frugivore interaction networks. Specifically, we address the following questions: (1) Is there a temporal turnover of bird species at the networks' core and periphery, associated with changes in the proportion of migratory bird species; and (2) Are the proportion of migratory bird species and the changes in food availability associated with temporal changes in the networks' parameters? The role of particular species in determining network structure may change temporally. Thus, it is necessary to know which species are essential for maintaining network stability and functionality. In this sense, migratory birds could play an important role during migration periods, because certain species may contribute to maintaining network functionality and resilience. Incorporation of interaction network analyses in conservation planning and management of tropical ecosystems is a promising approach, as it considers parameters relevant at the community level, which are in turn relevant for ecosystem function (*Valiente-Banuet et al.*, *2015*).

## METHODS

### Study area

This study was carried out in a seasonal tropical ecosystem in the Gulf of Mexico, at the "La Mancha" Coastal Research Center (CICOLMA), Veracruz, Mexico. Its coordinates are 19°35′25N and 96°22′49W, and its altitude ranges from 0 to 50 m asl. The climate is hot subhumid with summer rain of the Aw2 type, according to Köppen climate classification modified by *García* (*1981*). The minimum and maximum temperatures are 16 °C and 36 °C, respectively; mean annual temperature ranges between 22 and 26 °C, and mean annual precipitation ranges between 1,200 and 1,500 mm (*Gómez-Pompa*, *1972*; *García*, *1981*). Approximately 78% of the total precipitation takes place from June to September. Lower temperature conditions, scarce precipitation and maximum wind speeds are registered from November to February, when frequent cold fronts strike the area (*Castillo & Carabias*, *1982*). The CICOLMA comprises an area of approximately 83.29 ha where several plant communities occur (mangrove, savanna, tropical deciduous forest, flooded deciduous forest, sand dune scrub, and secondary forest <10 y); it is surrounded by farmland and cattle pastures (*Dubroeucq et al.*, *1992*; *Travieso*, *2000*).

More than 75% of the original natural vegetation cover in the region has been lost (*Portilla*, *1996*; *Ruelas, Hoffman & Goodrich*, *2005*). Nonetheless, vegetation heterogeneity at the landscape level is associated with high plant and animal richness at CICOLMA, which in turn promotes high diversity of plant–animal interactions (*Martínez, García-Franco & Rico-Gray*, *2006*). CICOLMA has been designated as an area of importance for bird conservation (AICA-02; *González-García & Ortiz-Pulido*, *1999*), and it is part of the main migration route of North American birds (*Thiollay & Nocedal*, *1978*; *Thiollay*, *1980*; *Straub & Martínez*, *2001*). Migratory bird abundance peaks during November–March, while the presence of some migrant species may occur during most of the year (*Ortíz-Pulido et al.*, *1995*). Around 299 bird species have been reported at CICOLMA, from which 161 are resident species, 102 are wintering migrants, 31 are transient migrants, and five are intertropical migrants (*González-García*, *2006*). It has been documented that approximately 108 species include fruit as part of their diet, although only 21 species are considered mostly frugivorous (*Ortíz-Pulido et al.*, *1995*). A study by *Ortíz-Pulido et al.* (*2000*) at CICOLMA registered 176 plant-frugivore interactions between 54 bird species and 33 plant species. It has been described, however, that there are more than 70 plant species whose seeds might be dispersed by birds and that up to 74 species of migratory birds may include fruits in their diet (*Ortíz-Pulido et al.*, *1995*), thus increasing the potential number of plant-frugivore interactions.

### Plant–bird interactions

Ten samplings were carried out during one year to record plant-frugivore interactions. Samplings took place in November 2013 (Nov13), January (Jan14), March (Mar14), May (May14), June (Jun14), July (Jul14), August (Aug14), September (Sep14), October (Oct14) and November 2014 (Nov14). All interactions between fruiting plants with endozoochorous syndrome and frugivorous birds were registered at four representative habitats at the study site (tropical deciduous forest, flooded deciduous forest, sand dune scrub, and secondary

forest >10 years). Because of the small size of the study site and short distance among habitats (<500 m), data for all habitats were pooled and are considered representative of CICOLMA. Plant-bird interactions were sampled with two complementary methods: mist nets and focal observations.

### Mist nets

During each of the ten sampling periods, each habitat was sampled by opening six mist nets (12 × 2.6 m) from sunrise to noon and from 16:00 h to sunset (total of eight hours). The total sampling effort was 1920 net hours. Each captured bird was kept at least 30 min in a cloth bag with a metal mesh bottom (to facilitate the collection of fecal samples).

### Focal observations

Ten samplings were also carried out in order to register interactions by direct observation. Each sampling consisted of visual inspection along one transect per habitat, carried out from sunrise to 11:00 h and from 16:00 h until sunset (6.2 h ± 0.91). Transect length varied according to habitat extension: tropical deciduous forest, 1,338 m; flooded deciduous forest, 511 m; sand dune scrub, 808 m; and secondary forest, 1,595 m. Observations focused on fruiting plants with endozoochorous syndrome and had a duration of 10–15 min per plant species. The total sampling effort was 280 h of observations.

### Seed identification

The identification of fleshy fruits (focal observations) and seeds (fecal samples) was carried out through the construction of a reference guide. For that purpose, fruits from all plant species were collected along the transects used for focal observations (see above). The reference guide contained seeds, photographs of ripe fruit, information on seed number per fruit, and the species life form. Seeds not included in the reference guide were identified with the help of other collections (Instituto de Ecología, A.C.; Instituto de Biotecnología y Ecología Aplicada at Universidad Veracruzana).

We considered an interaction to have occurred when we observed a bird species swallowing fruits (focal observations) and/or we found seeds in a bird's defecation (mist nets). The number of fruits consumed was either obtained directly through focal observations or indirectly through the fecal samples. In the latter case, we estimated the number of fruits consumed per bird by counting the number of seeds in the defecations and dividing this number by the known mean number of seeds per fruit for that particular plant species.

## Explanatory variables
### Fruit availability

The number of species and fruiting individuals were quantified along the same transects where focal observations were performed by using an 8-m transect width. The total sampled area, considering all vegetation types, was 1.7 hectares (0.017 sq. km). Each fruiting individual was assigned a value of ripe fruit abundance index (FAI) (*Saracco, Collazo & Groom, 2004*) according to the following ordinal scale: 1 = 1–10 fruits; 2 = 11–100 fruits; 3 = 101–1,000 fruits; 4 = 1,001–10,000 fruits; and 5 > 10,000 fruits. Species

richness of fruiting plants was calculated as the total number of species bearing ripe fruits in a given sampling period. For fruit abundance, we obtained the average of the FAI categories for each plant species in each sampling period.

### *Proportion of migratory species*

Based on all the bird species registered eating fruits, we calculated the proportion of migratory species in each of the ten samples. Each bird species was classified as migratory or resident based on the Cornell Lab of Ornithology listing of Neotropical birds (*Schulenberg*, *2015*) (http://neotropical.birds.cornell.edu/portal/species/overview) and with the help of expert ornithologists in our study area. For the purposes of our study, which was to assess the effects of all bird species that are not year-round residents, we included in our "migratory" category both stopover species and winter residents.

## Response variables

Two kinds of interaction matrices were built for each of the ten samples: a qualitative matrix (presence/absence data) and a quantitative matrix (interaction intensity data). The interaction intensity between a plant species and a bird species was given by the total estimated number of fruits eaten (*García et al.*, *2014*). We chose this metric as a proxy for interaction intensity because we were interested in discussing our results in terms of the potential effectiveness of seed dispersal by birds; in this sense, we believe that the number of fruits is a better proxy of the quantity component of dispersal effectiveness. The number of fruits eaten was obtained by adding the direct count of fruits obtained during focal observations and the extrapolated number of fruits from fecal samples (see above).

## Data analyses

### *Network-level parameters*

In order to characterize the temporal variation in network parameter values, the following parameters were calculated for each sampling period: network size, connectance (C), nestedness (NODF), network-level specialization ($H_2$), interaction strength asymmetry (ISA) and bird niche overlap (NO).

*Network size* refers to the total number of frugivorous bird species and fruiting plant species making up the network. *Connectance* is the proportion of registered interactions as compared to the total possible interactions given the observed species. *Nestedness* is the network pattern where specialist species interact only with generalist species, but the latter also interact among themselves (*Bascompte et al.*, *2003*). Nestedness was calculated with the NODF estimator using ANINHADO 3.0.2 software (*Guimarães & Guimarães*, *2006*). As a means of assessing the statistical significance of parameter estimates, NODF values were compared to nestedness values for each one of the 1,000 network replicates generated randomly while considering the observed richness of species and interaction heterogeneity ("null model 2," according to *Bascompte et al.*, *2003*). *Network-level specialization* ($H_2$) is a measure of niche segregation between species based on the deviation between the real number of species interactions and the total number of expected interactions for a network. This index assumes that all species interact with their mutualistic species at the observed rates. Given that this metric is not affected either by sampling effort or by network size, it

allows for robust and reliable comparisons among networks (*Blüthgen, Menzel & Blüthgen*, *2006*). $H_2$ values range from 0 (no specialization) to 1 (perfect specialization for a given total number of interactions). *Interaction strength asymmetry* measures the difference between birds dependence on plants vs. plants dependence on birds. *Niche overlap* is a measure of similarity for the interaction pattern between the species within each interacting group. All network parameters (except nestedness, as seen above) were calculated with 'bipartite 2.05' (*Dormann et al.*, *2009*) using the R software v.3.1.2 (*R Development Core Team*, *2012*).

In order to determine which species comprised the network core and which ones comprised the network periphery, a core–periphery analysis was performed using the following formula:

$$Gc = \left( \frac{k_i - \sigma k_{\mathrm{mean}}}{\sigma_k} \right),$$

where $k_i$ is the average number of links maintained by the *i*th frugivorous species, $k_{mean}$ is the average number of links maintained by all the frugivores in the network, and $\sigma_k$ is the standard deviation of the number of links maintained by the frugivorous species in the network (*Dáttilo, Guimarães & Izzo*, *2013*). Values of $Gc < 1$ correspond to species with a small number of interactions that are part of the network periphery, whereas $Gc > 1$ values correspond to species with a large number of interactions and comprise the generalist core.

## Statistical analyses

To evaluate whether periods of high and low proportion of migratory frugivorous bird species were significantly associated with a temporal turnover of bird species at the network core and its periphery, we used a PERMANOVA (Permutational Multivariate Analysis of Variance) (*Anderson*, *2005*), followed by a graphical representation of the results through a non-metric multidimensional scaling (NMDS) ordination. To carry out the analysis and ordination, the ten samplings were *a priori* grouped into two categories: (i) high proportion of migratory bird species: Jan14, Mar14, Sep14 and Oct14; and, low proportion of migratory birds species: Nov13, May14, Jun14, Jul14, Aug 14 and Nov14. The networks core and periphery were analyzed separately. The PERMANOVA was performed by means of the *adonis* function from R's 'vegan' package (*Oksanen et al.*, *2012*). This function uses the Bray–Curtis index and implements a multivariate variance analysis using the distance between matrices (species composition dissimilarities in each sampling), from which probabilistic significance is obtained through 999 permutations (*Anderson*, *2005*).

We utilized generalized linear models (GLM's) to find out the extent to which temporary variations in the proportion of migratory species and resource availability influenced plant-frugivore network parameters. Given that the proportion of migratory species and fruit abundance were negatively correlated (Spearman correlation, $r_s = -0.79$, $P \leq 0.01$), two types of models were explored: (1) relating proportion of migratory species, fruit richness, and their interaction, with network parameters, and (2) relating fruit abundance with network parameters. Poisson error distribution and log link function were used. Data underdispersion ($\varphi < 1$) and overdispersion ($\varphi > 1$) cases were dealt with by using "quasi-Poisson errors" (*Crawley*, *2007*). All statistical analyses were conducted using R v. 3.2.1.

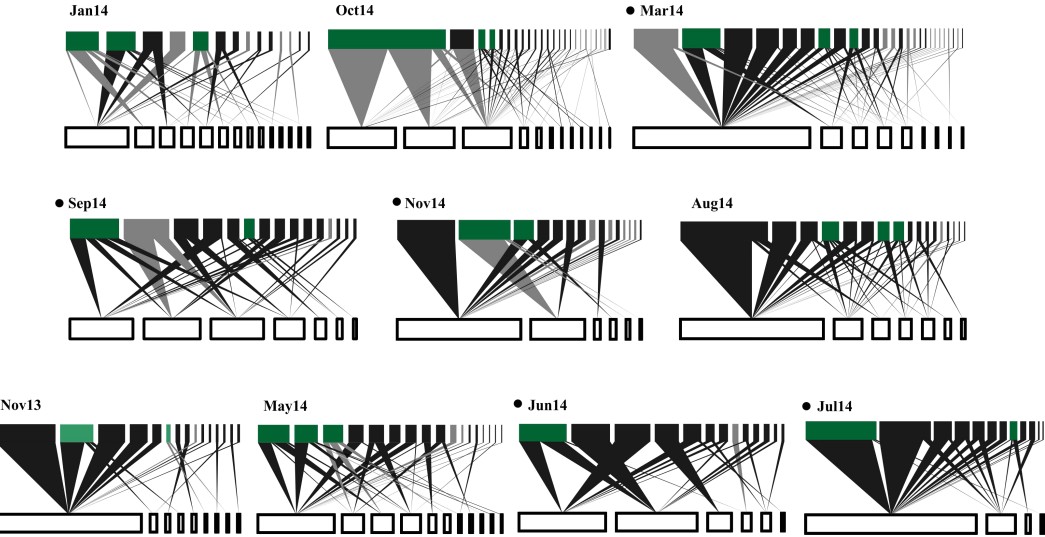

**Figure 1** **Temporal variations in plant-frugivore network architecture at CICOLMA.** Networks are ordered (left to right and upper to lower) from high to low proportion of migratory species present. In each network, bars at the bottom represent plant species (white nodes) and bars at the top represent frugivorous bird species (black nodes for resident species and gray nodes for migratory species); green nodes represent species (migratory or resident) belonging to the generalist core ($Gc$). The link width represents interaction intensity. Black points nest to the newtork label indicate significantly nested networks ($P \leq 0.05$). For names of plant and bird species see Table S1.

# RESULTS

A total of 319 plant-bird interactions were registered, including 42 plant species and 44 bird species. Out of the ten networks obtained in our year-long study, only five exhibited significant nestedness (Fig. 1; Table S1). The proportion of migratory bird species peaked in Jan14 and Oct14, whereas no migratory birds were registered in Jun14 (Fig. 2A). Fruit abundance peaked in Jun14 and Jul 14 (Fig. 2B), while fruit richness was greater in Jan14 and Sep14 samplings (Fig. 2C). The largest networks, also holding the largest number of interactions, were registered in Mar14 and Sep14 (Fig. 2D). The most connected network with the highest nestedness values occurred in Jun14, (Figs. 2E and 2F). The highest value of interaction strength asymmetry was recorded in Oct14 (Fig. 2G), and birds' niche overlap was highest in Mar14, Jun14 and Oct14 (Fig. 2H). The less specialized networks occurred in Mar 14 and Jun 14, and the highest specialization values were observed in May14 and Oct14 (Fig. 2I).

## Temporal turnover in network core and periphery bird species composition

The network core did not showed a turnover in species composition between periods of high and low proportion of migratory species (PERMANOVA, Pseudo-$F = 0.93$; $p = 0.52$; (Figs. S1A and S1B). By contrast species composition in the networks periphery was significantly dissimilar between periods with high and low proportion of migratory bird species (PERMANOVA, Pseudo-$F = 2.00$; $p = 0.01$).

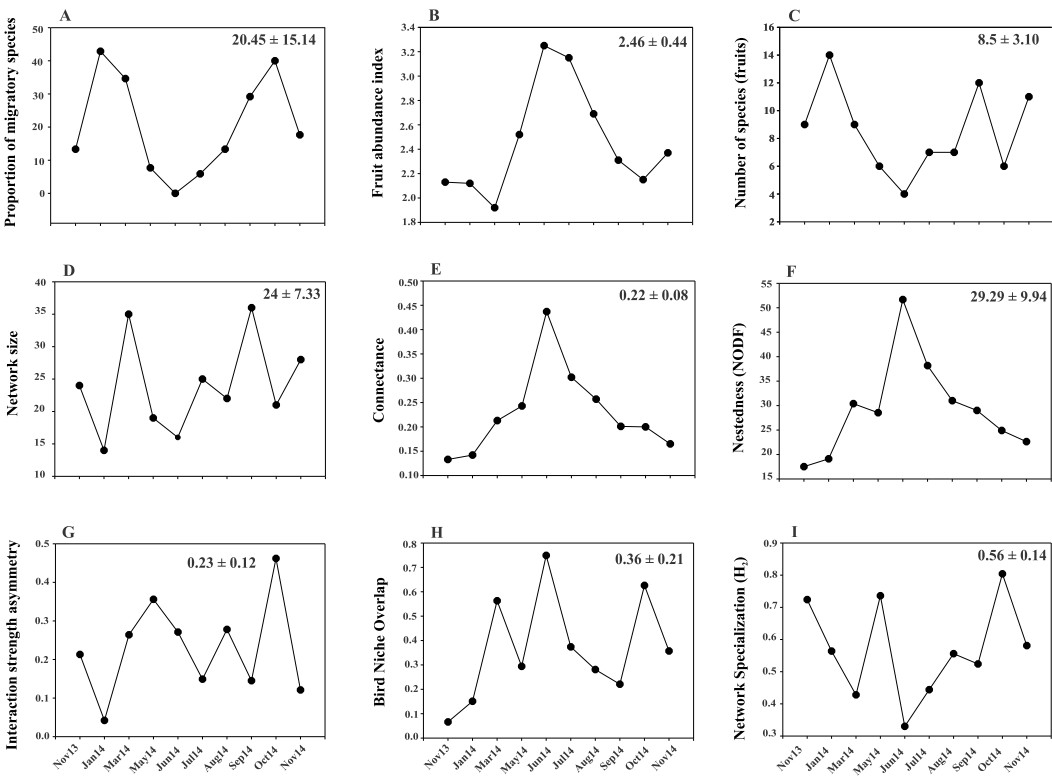

**Figure 2** **Temporal variability in the values of proportion of migratory species.** (A), fruit richness and abundance (B, C) and network-level parameters: network size (D), connectance (E), nestedness (F), interaction strength asymmetry (G), bird niche overlap (H) and network specialization (I). Values in the upper right corner of each panel represent the mean ± SD.

In general, resident birds were always part of the networks generalist core (*Gc*), whereas migratory birds were represented in the networks generalist core in five of the ten samplings. Seven resident bird species formed the network core (*Melanerpes aurifrons*, *Pitangus sulphuratus* and *Psarocolius montezuma* being the most common), but only three migratory bird species were part of the core (*Dumetella carolinensis*, *Vireo griseus* and *Tyrannus tyrannus*; Fig. 1). Lastly, resident bird species were predominant at the networks periphery (Fig. 1).

## Influence of proportion of migratory species and food availability on network parameters

Network size revealed a positive and significant relationship with the proportion of migratory bird species and with fruit species richness (Figs. 3A and 3B; Table S2A). Connectance was negatively and significantly related to the proportion of migratory birds and fruit richness (Figs. 3C and 3D; Table S2B), and positively related with fruit abundance (Fig. 3E; Table S2B). Additionally, nestedness revealed a negative relationship with the proportion of migratory species and with fruit richness (Figs. 3F and 3G; Table S2C), but a positive one with fruit abundance (Fig. 3H). Interaction strength asymmetry was positively and significantly related only to the proportion of migratory species (Fig. 3I; Table S2E). Network-level specialization ($H_2$) and niche overlap was not significantly

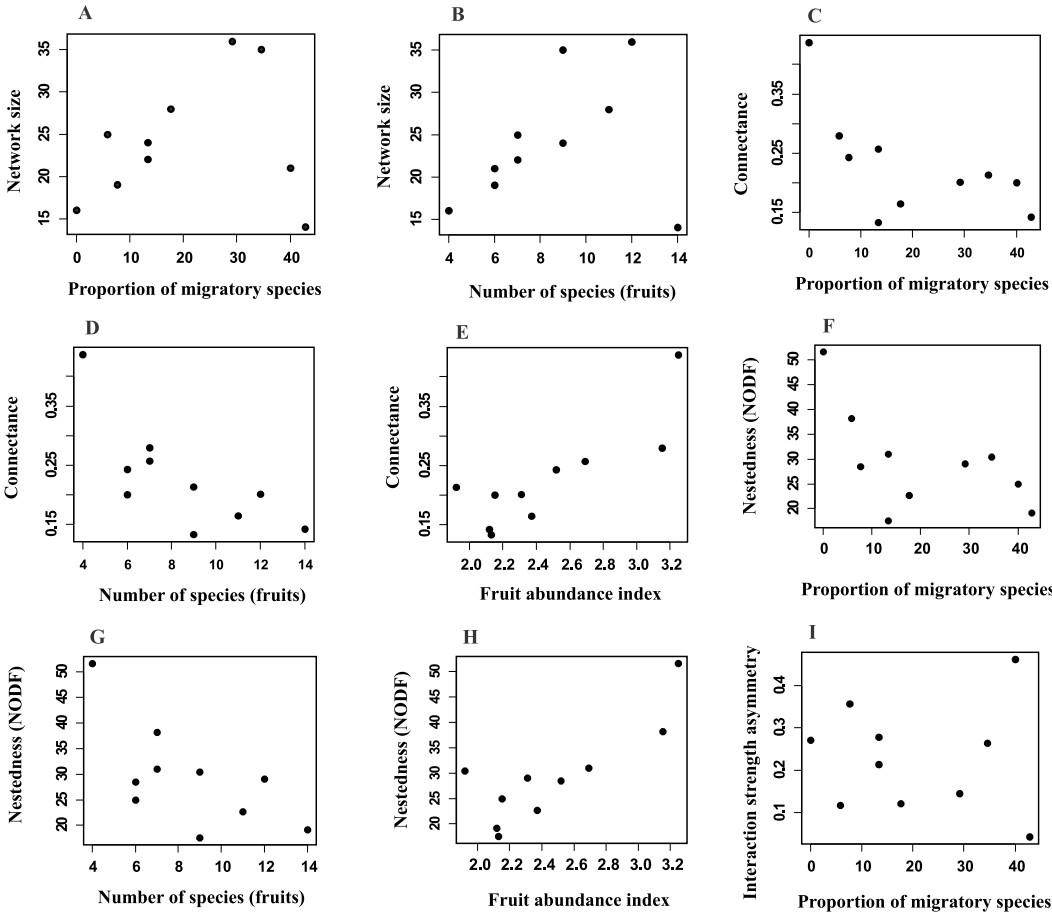

**Figure 3** **Relationships between network parameters and explanatory variables proportion of migratory species, fruit richness and fruit abundance index.** Network size (A, B), connectance (C–E), nestedness (F–H) and interaction strength asymmetry (I). Only significant ($P \leq 0.05$) relationships are shown. See Table S2.

associated with any explanatory variable considered (Tables S2D and S2F). The interaction term between the proportion of migratory bird species and fruit richness had a significant effect on network size (Table S2A), which indicated that larger-sized networks occurred during periods in which the proportion of migratory species was higher but the richness of fruiting species was lower. On the other hand, connectance, nestedness, specialization, interaction strength asymmetry and niche overlap were not affected by this interaction (Table S2).

## DISCUSSION

Many studies have demonstrated how phylogenetic signals (*Rezende et al., 2007*), habitat disturbance (*Nielsen & Totland, 2014*), trait matching (*Stang, Klinkhamer & Meijden, 2006*; *Blüthgen et al., 2007*; *Chamberlain & Holland, 2009*; *Stang et al., 2009*), biological invasions and fruit abundance (*Krishna et al., 2008*; *Vázquez et al., 2009*) can affect the structural parameters of plant-frugivore interaction networks. Earlier studies also suggested that changes in other parameters of fruit availability, as well as the presence of migratory

species, might affect the interaction between plants and frugivorous birds (*Karr*, *1976*; *Rey*, *1995*; *Jordano*, *1993*; *Jordano*, *1994*; *García, Zamora & Amico*, *2011*). Those studies, however, did not use network analysis to assess the importance of these variables at the community level. This constitutes one of the main contributions of our research. Our study further highlights a strong temporal turnover of bird species in the periphery of the network, as well as the importance of including temporal dynamics of plant-frugivore networks, particularly in highly seasonal tropical forests.

## Temporal turnover in bird species composition of network core and periphery

Our analysis showed that the proportion of migratory species was not related to species turnover in the networks core. However, there was a significant turnover of species in the network's periphery when comparing periods of higher and lower proportion of migratory species. Thus, core species, compared to peripheral species, displayed less temporal turnover (*Bascompte et al.*, *2003*; *Dáttilo, Guimarães & Izzo*, *2013*). Such a pattern could be caused by the abundance of core species, given that the most abundant plant and animal species tend to interact among themselves more than the less connected species, thus contributing to network core stability (*Vázquez et al.*, *2007*; *Vázquez et al.*, *2009*).

Additionally to abundance, bird behavior (e.g., competitively superior and/or aggressive species) may also be a determining attribute of species composition in the networks core (*Dáttilo, Guimarães & Izzo*, *2013*). For instance, the migratory birds constituting the core of the network in our study site, *Vireo griseus*, *Dumetella carolinensis* and *Tyrannus tyrannus*, are generalist species (i.e., they have many links in the network). Their incorporation into the system is comparable to that observed in invading species, which have been shown to quickly become an integral part of the generalist core of interaction networks (*Traveset & Richardson*, *2006*; *Aizen, Morales & Morales*, *2008*; *Díaz-Castelazo et al.*, *2010*). It should be mentioned that the network core was not exclusively composed of migratory birds, but also of larger resident bird species such as *Melanerpes aurifrons*, *Pitangus sulphuratus* and *Psarocolius montezuma*, species quite abundant at the study site (M Ramos, pers. obs., 2014) and displaying territorial behavior (*Fitzpatrick*, *1980*; *Husak*, *2000*; *Price, Earnshaw & Webster*, *2006*).

The relevance of core species is that they may contribute to maintaining network stability by making them more resilient against a variety of disturbances, through an increase in network robustness and cohesion (*Bascompte et al.*, *2003*; *Díaz-Castelazo et al.*, *2010*; *García, Zamora & Amico*, *2011*; *Chama et al.*, *2013*; *Nielsen & Totland*, *2014*; *Vidal et al.*, *2014*). Also, it is possible that core species are competitively superior (*Dáttilo, Guimarães & Izzo*, *2013*) and may be more effective as seed dispersers, exerting selective pressure on some fleshy-fruited plant attributes (*Ruggera et al.*, *2015*).

Most birds comprising the networks periphery were resident species (27 resident species *vs*. 17 migratory species), which could be a consequence of these species maintaining more exclusive interactions with plants (*Ruggera et al.*, *2015*). In addition, peripheral species could have less territorial behaviors, which could lead to displacement by generalist species from the core (*Bascompte et al.*, *2003*; *Dáttilo, Díaz-Castelazo & Rico-Gray*, *2014*).

## Influence of proportion of migratory species and food availability on network parameters

Most studies on plant-frugivore networks have used a static approach, describing networks as snapshots in time. Our research demonstrates, though, that plant-frugivore interaction patterns are temporally dynamic and are significantly influenced by factors such as seasonal variation in food availability and the incorporation of new frugivore species into the system.

### Fruit abundance

Our data showed that, within a year, fruit abundance was positively associated with network connectance and nestedness. This result suggests that during periods high fruit abundance, networks are more complex and more robust to secondary extinctions (*Dupont, Hansen & Olesen*, *2003*; *Heleno, Devoto & Pocock*, *2012*). It is likely that the nested pattern is mostly driven by highly connected plant species with high fruit abundance (e.g., *Bursera simaruba* and *Ficus* spp.), which allows them to maintain interactions with a large number of bird species. The importance of these key plant species to our study site lies in their maintaining numerous generalist interactions, because they provide a continuous food supply for birds. In addition, these species are probably the main contributors to network nestedness and therefore to its robustness against disturbance (*Olesen et al.*, *2007*; *Tylianakis et al.*, *2010 Ruggera et al.*, *2015*).

Interestingly, we found a negative correlation between the proportion of migrant species and fruit abundance. This is somewhat contrary to the results of previous studies in which a positive relationship between food availability and the abundance of particular migratory bird species was recorded (*Thompson & Willson*, *1979*; *Loiselle & Blake*, *1991*; *Loiselle & Blake*, *1992*; *Jordano*, *1994*; *Rey*, *1995*; *Herrera*, *1998*; *García, Zamora & Amico*, *2011*; *Guitián & Bermejo*, *2006*; *Mulwa et al.*, *2013*). In our study site, however, this negative relationship could have occurred because the periods of lower fruit abundance coincided with periods of greater fruit richness, involving a higher proportion of migratory species during these periods. In addition, studies reporting positive relationships between the abundance of migrants and the abundance of fruit were mostly carried out in more homogeneous landscapes whereas our study site features a mosaic of different vegetation types. Such habitat heterogeneity is probably associated with the high diversity in fruits, which in turn could favor migratory bird species, which can be highly dependent on fruit during migration (*Loiselle*, *1987*).

### Fruit richness

As already mentioned above, the periods with highest richness of fruiting species coincided with periods of higher proportions of migratory species. The positive relationship between fruit richness and proportion of migratory species was in turn associated with larger networks, but with lower values of connectance and nestedness. It is probable that an increase in network size may have exerted an influence on network parameters, decreasing connectance and becoming less cohesive.

Also, there was a negative relationship between fruit abundance and fruit richness. We believe that this pattern might emerge from the fact that plant species with different

fruiting phenology may also differ in their crop sizes. It seems that in our study site, fruit availability during the dry season was dominated by a few plant species, mostly trees, with very large fruit crops (e.g., *Bursera*, *Ficus*). Conversely, during the rainy season, more plant species fruited, but with smaller fruit crops (e.g., lianas, shrubs).

It is known that those networks possessing a higher richness of interacting species compared to interaction-poor communities, tend to be more nested, and therefore, they are more stable and more robust when facing disturbances (*Wright & Reeves*, *1992*; *Ulrich*, *2009*; *Atmar & Patterson*, *1993*). Nonetheless, and contrary to this, we discovered a negative relationship between nestedness, and fruiting plant richness and proportion of migratory birds on the other hand. These unexpected relationships could be ascribed to higher food richness periods and "new" species in the system favoring more selective plant-bird interactions (with fewer links), rendering less connected and nested networks, as has been suggested by other studies (*Joppa et al.*, *2010*).

On the other hand, high fruit richness probably influenced resource use diversification by frugivorous birds through the reduction of niche overlap in our study system (*Díaz-Castelazo et al.*, *2013*). In fact, other authors have reported that fruit richness may decrease interspecific competition between frugivorous bird species (*Blüthgen et al.*, *2007*; *Chama et al.*, *2013*), thus reducing diet overlap (*Vázquez et al.*, *2007*; *Vázquez et al.*, *2009*).

The presence of migratory frugivorous birds in the system "disordered" the plant-frugivore network organization by reducing the nested pattern (*Bascompte et al.*, *2003*). Also, network disorder was evident through changes in the interaction strength asymmetry, which was positively related to the proportion of migratory bird species, which could in turn be caused by temporal variations in the abundance of particular bird species (*Carciner, Jordano & Melian*, *2009*). In other words, periods with the highest proportions of migratory species were also the ones with greatest bird abundance (e.g., more than 300 individuals from one species associated with one plant species individual; M Ramos, pers. obs., 2014). During such periods, it is possible that birds depend more on plants, than vice versa.

Other work has shown that the incorporation of many new species into a system (e.g., invasive species) increases interaction asymmetry (*Tylianakis et al.*, *2008*), which originates increased interaction diversity and robustness against disturbances (*Bascompte, Jordano & Olesen*, *2006*). This could have significant ecological implications at the community level, given that the most abundant species tend to exert a stronger influence on the species they interact with. This asymmetry of interaction is frequent in nested networks, where species with few connections interact more with the highly connected ones. Nevertheless, *González-Castro et al.* (*2012*) suggested that interaction strength asymmetry is not necessarily given exclusively by the abundance of species, but may also be influenced by other factors such as plant phenology and species-specific characteristics.

Previous studies have shown that mutualistic networks tend to have low specialization levels (*Jordano*, *1987*; *Blüthgen et al.*, *2007*; *Schleuning et al.*, *2011*; *Chama et al.*, *2013*;), and it is possible that this characteristic may contribute to network persistence and robustness in case of species extinction (*Bascompte & Jordano*, *2007*). The temporal variations of interaction specialization in our plant-frugivore network, produced values from low to high (0.3–0.8), and such specialization changes were not explained by variability in resource

availability or the proportion of migratory species. It is probable that other variables not quantified in the present study, such as fruit attributes (size, weight, color, etc.), as well as bird characteristics (size, foraging strategies, etc.) could be related to the temporal variations in our systems specialization.

Although our study did not evaluate the effectiveness of seed dispersal, our results have implications in this regard. As already mentioned, the nested structure of the network, its size, connectance and interaction asymmetry, were related to the temporal dynamics migratory species and fruit availability. We also mentioned that a nested pattern gives stability to the network by making it more resistant against secondary extinctions. This pattern can be lost during certain periods of time, however, due to temporal dynamics as those described in this study. On the other hand, larger networks that have low connectance, could favor the effectiveness of seed dispersal for the plant species, because its seeds would be dispersed by a greater diversity of interacting bird species (*Olesen & Jordano*, *2002*). In addition, the interaction strength asymmetry in our study, related to the proportion of migratory species, could promote seed dispersal between habitat patches, due to the high consumption of large patchy fruit crops by some species of migratory birds (*Jordano*, *1982*).

To conclude, studies as the present one, showing the temporal dynamics of network parameters could be essential for conservation and management purposes. Fluctuations in resource availability and bird species composition are prone to be affected by anthropogenic disturbances, with cascading effects on network structure and important consequences for ecosystem function.

## ACKNOWLEDGEMENTS

We especially wish to acknowledge Antonio Lopez-Carretero for logistical support in the field and data analysis. We would also like to thank Enrique Romero for his support during fieldwork and the personnel working at CICOLMA for all their support.

### Funding
The authors received partial funding from the following grants and institutions: Grant number 234062 for M Ramos-Robles by CONACYT, Project number 2010-152884 SEP-CONACYT for EA and Project number 2003011143 for C Díaz by Instituto de Ecología, A.C. (INECOL). The funders had no role in study design, data collection and analysis, decision to publish, or preparation of the manuscript.

### Grant Disclosures
The following grant information was disclosed by the authors:
CONACYT: 234062.
EA: 2010-152884 SEPCONACYT.
Instituto de Ecología, A.C. (INECOL): 2003011143.

### Competing Interests
The authors declare there are no competing interests.
## Author Contributions

- Michelle Ramos-Robles conceived and designed the experiments, performed the experiments, analyzed the data, contributed reagents/materials/analysis tools, wrote the paper, prepared figures and/or tables, reviewed drafts of the paper.
- Ellen Andresen and Cecilia Díaz-Castelazo conceived and designed the experiments, contributed reagents/materials/analysis tools, wrote the paper, reviewed drafts of the paper.

## Data Availability

Raw data has been uploaded as Supplemental Information.

## Supplemental Information

Supplemental information for this article can be found online at http://dx.doi.org/10.7717/peerj.2048#supplemental-information.

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
