# Peer review of "Temporal changes in the structure of a plant-frugivore network are influenced by bird migration and fruit availability"

_PeerJ, doi:10.7717/peerj.2048_

## Round 0.1 · original submission · Major Revisions

· Academic Editor

Major Revisions

Dear aithors
many thanks for submitting your ms to our journal. As you can see our reviewers found several issues which need a thorough revision. Please accept the criticism and revise your ms accordingly.
Kind regards

Michael Wink

Reviewer 1 ·

Basic reporting

The paper needs to be revised in terms of its English.

Experimental design

The design of the study (field work) is straight forward. Data analysis is extensive and follows the modern network concept.

Validity of the findings

The findings are described sufficiently well. In terms of ecology, the results are hardly surprising when one takes all the data analytical accessories away.

Additional comments

This study describes the dynamics of plant – avian frugivores interactions using network analyses. Network analyses are powerful tools to describe the structure of species interactions, but are of little explanatory value unless coupled with functional investigations (e.g. Ruggera; see also the critical review by Tylianakis et al. 2010). The authors are right when they state that the challenge is to determine the processes that are involved in the configuration of the interactions (line 42). Letting these general problems aside, the paper provides valuable data on the seasonal dynamics of seed dispersal in a Neotropical landscape with a special emphasis on the role of migratory birds. Its readability, however, partially suffers from language problems. The literature cited is extensive, pertains mainly to network theory and statistical methods, ignoring largely earlier work on frugivory and seed dispersal.
Lines 62-63: How can migrants change plant abundance and species composition? What exactly is meant here, and what is the time scale referred to?
I cannot follow the reasoning behind the statement on lines 67-69. Is “arrival” meant to refer to those species that stay in the area over winter? Throughout the paper, “migrants” seems to refer to both stop-over and overwintering birds. “Arrival” is also used on line 78. Staying on this line: That this study is the first that considers the role of frugivorous migrants would be overstated when considering studies that do not use network analyses (e.g. Leck 1972, Karr 1976, Howe & De Steven 1979, Thompson & Willson 1979, Jordano 1982, Loiselle 1987). Here, and in the paper in general, the methodological aspect is clearly overemphasized, pushing ecological and evolutionary aspects into the background.
When specifying the questions (lines 80-83) it would be advisable to somehow specify why these questions are important beyond finding changes in some descriptive parameters. What would be the theoretical expectations regarding migrants and their functional role?
The terms low and high migration periods, as conceived here (lines 219-225) suffer from the already mentioned problem that the proportion of stop-over migrants and winter residents are both taken as representing “migration”.
Migration rate (lines 158-159): A proportion like this does by no means constitute a rate. The usage of this term is clearly misleading and is related to the problem of dealing with migrants and winter residents already alluded to.
Important network parameters are defined on lines 177-182, but were used already earlier in the paper. It may be helpful to refer to these definitions upon mentioning the terms, or define them right at their first appearance.
The negative correlation between proportion of migrants and fruit availability is somewhat surprising and interesting (Rey 1995, Guitián & Bermejo 2006, Mulwa et al. 2013; but cf. Loiselle 1987). It should be further discussed in biological/functional terms, rather than with another formal data analysis exercise.

Line 257: “no lack” renders the statement incomprehensible. And the whole analysis in this paragraph is confounded by the way “migration” has been defined in this paper. All the migratory core species are winter residents (263-264).
The discussion dwells on the hardly surprising fact that “bird migration” (in quotes because of the peculiar usage of the term in this study) changes the interaction network. In line 305 and what follows, the verb “integrate” is used in its transitive form. Thus it implies processes that are hard to believe to occur in that system (By what means would migrants integrate?). There seems to be another language problem on line 315. I would have theoretical/philosophical reservations concerning the statement on line 320.

Reviewer 2 ·

Basic reporting

Many previous studies showed that fruit availability and bird migration affect seed dispersal and thus the main results of the current study "we showed that bird migration and resource availability influence the temporal structuring of plant-frugirovre bird network" (line 290-1) is not a big surprise. The main novelty here is the use of network analysis to show the dynamics of fruit producing-plants and their fruit consumers.

The article should be read and edit by native English speaker.
All figures are poorly presented and need a major revision. The text in the figures is too small and unclear. Please add units in the Y axis as appropriate. Keep the figure order as in the text in the result section. E.G. Fig. 1a should precede Fig. 1b in the text.

Experimental design

I find the two research questions well defined, relevant & meaningful. However, I have many reservations with regard to the methodology:
Although there were four different types of habitats, the authors decided to pool the results obtain in these habitats. This decision should be explained. I assume that both bird and fruiting plant species are different among these four types of habitats. E.g., in sand dune scrub one supposed to see a totally different plant communities as well as huge different in bird presence. Therefore, I don’t understand the ecological logic behind the decision to pool them together. On the other hand, habitat type could be used as a factor in the analysis and demonstrate network similarities among the four habitats.

Focal observations: how the author verify that the birds that were "foraging on the fruits" (line 147) actually consumed the fruits and not insects that are inhabited the fruit surface? Furthermore, bird consumers are not necessary seed dispersers but may be seed predators.

More details are needed on how fruiting plant species richness was calculated.
The authors do not show any data on bird richness, bird abundance and bird diversity and their effect on the network.

The presentation of any average that was calculated in the study should be presented in the text and in the figures with its standard deviation. E.g., FAI, fruiting richness (lines 156-7).

The calculation of the interaction intensity (lines 166-7) should be better explained. Why the "total registered number of fruits eaten" is the true measurement of the interaction intensity? For example, I understand that the same interaction intensity will be calculated in these two situations: 100 fruits of plant species A that were consumed from one individual plant versus 100 fruits of plant species A that were consumed from 20 different plants. Furthermore, it is not clear how the number of consumed fruits were calculated based on two different methodologies: direct fruit count and seeds that were obtained in fecal samples (lines 168-9). The author should clarify this calculation.

The calculated Gc (lines 199-208) is mentioned only in the Methods but not in the Results/Tables/Figures.

Fig. 1 The use of 1,2,3 in the x-axis to represent November, January, February is misleading. Use 1 for January, 2 for February etc. This is true throughout the whole manuscript. Other option: Jan13, Feb13….Nov14.

"Fruit abundance remained relatively constant" (line 240) – but in Fig. 1 there are dramatic differences: C1 ~ 2.1, C5 - ~ 3.2, this is X1.5 and not constant.
"fruit richness was greater in January and September sampling" (line 241): an explanation is needed for how fruit richness was calculated.

"out of the ten networks….only five exhibited significant nestedness (P<0.05; Figure 1)" (lines 244-5): I don’t see in Fig. 1 which network is significant.

NMDS analysis: The purpose of this analysis is unclear. It is also unclear how this analysis was conducted. What was the data that was used in this analysis? The description of Fig. 2 is insufficient as well. No details on the methodology, e.g., did they use the Bray-Curtis dissimilarity index? (are lines 215-218 applied for the NMDS as well? If so, please write it). The authors should explain very well what we can get from this figure. Proportion (%) of variance explained by each axis should be shown. What is the main reason for the separation we see of bird species on the first and on the second axes? Both figures (2a, 2b) show a low stress values. What is the conclusion from this? Fig. 2b is not mentioned at all in the text of the result section. The authors should check the possibility to add ellipsoids that represent a 95% (or other %) confidence interval surrounding each group of bird species that are different.

Fig. 3 – The order (C2, C9, etc.) make no sense. There are no asterisks (*) to highlights significantly nested networks in the figure.

Fig. 4 – This figure should be redrawn. The text on the Y axes is not in English. The order of the figures is not clear. The actual correlations (rs?) and p value should appear on each figure. Many of the results in Fig. 4 are not mentioned in the text (e.g., Fig. 4d, 4e, 4f, 4g, 4h, 4i). Therefore the author should delete it.

Validity of the findings

Please explain how Tukey's test indicates that the species turnover at the periphery is significantly higher between migration periods than within them (lines 256-259).

Table S1: The authors were not identified 13 fruit-producing plant species which are 30% of the total plant species in this study. If these seeds were collected from droppings they may belong to plants that produce seeds without fruits. This information is important and should be mentioned in the Method section – how do the author identify the fruit species? How could they ensure that these were seeds from fruits? This validation is necessary.

The negative correlation between fruit richness and fruit abundance (Fig. 4f) should be explained.

Table S2c: P value of 0.052 is not significant and thus should not be in bold. Hence, lines 278-280 should be revised.

Additional comments

I suggest that the authors will add a short paragraph into the Discussion in which they explain how their main findings (e.g., network size, connectance, nestedness, asymmetry) may affect seed dispersal efficiency.

---

## Round 0.2 · accepted · Accept

· Academic Editor

Accept

Dear authors,

Thanks for the revision which is now accepted. Thank you for submitting your work to our journal.

Regards
Michael Wink
Academic editor

Reviewer 2 ·

Basic reporting

I believe the authors have effectively responded to all the concerns I raised in my first review. My recommendation is to accept the manuscript for publication.

Experimental design

OK

Validity of the findings

OK